# Investigating Token-Level Supervision of Multi-Dimensional Attribute Combinations

## Abstract

In multi-dimensional attribute combination training for LLMs, the dimension conflict is an unavoidable issue. Since each token can have different influences on different dimensions, applying token-level supervision across multiple dimensions is a potential method to mitigate dimension conflicts. However, the difficulty in obtaining token-level supervision signals across multiple dimensions through annotation has hindered further investigation into supervision methods. In this work, we experimentally validate the impact of dimension conflicts on LLM training and propose a method for applying token-level supervision for multi-dimensional attribute combination training. This method establishes token-level connections between the trained model and attribute models using token sequences generated by the trained model for optimization, and controls the optimization process through entropy-based weight calculation, without requiring any additional token-level annotations or external models. This method effectively improves multi-dimensional performance and provides new insights into the investigation of token-level supervision for multi-dimensional attribute combinations.

## 1 Introduction

The evaluation of responses from large language models (LLMs) typically depends on a combination of attributes across multiple dimensions, such as instruction-following, helpfulness, truthfulness, honesty, etc. There are two ways to consider multi-dimensional attributes in LLM response quality scoring datasets: (1) explicitly consider multiple dimensions, score each dimension separately, and a weighted average comprehensive score can be obtained (Wang et al., 2024c); (2) directly provide a comprehensive score that implicitly considers multiple dimensions (Wang et al., 2025b). Whether considering multiple dimensions of attributes explicitly or implicitly, the supervisory signals ultimately used to train LLMs are usually sample-level scores.

When training LLMs using sample-level overall scores as supervision signals, considering combinations of attributes across multiple dimensions inevitably leads to dimension conflicts (Li et al., 2025b). For example, when applying direct preference optimization, for two responses to the same prompt, the overall quality score may be consistent with the preferences of certain attribute dimensions and inconsistent with those of other attribute dimensions. In this case, training on this pair of responses may have a negative impact on attribute dimensions with inconsistent preferences. One major issue with this conflict is that the overall quality score is used uniformly as a supervisory signal for each token, without considering that a token may have different effects on different attribute dimensions. One possible way to mitigate conflicts is to use token-level supervision signals (Abdin et al., 2024) to assign different optimization directions to different dimensions for each token. However, the main obstacle to implementing such token-level supervision is that token-level supervision signals for multiple attribute dimensions are difficult to obtain through direct annotations. To address this issue, we need to investigate token-level supervision methods that do not require additional token-level annotations for multi-dimensional attribute combinations. In this paper, we conduct our investigation according to the following outline:

- We demonstrate the dimension conflict issue through data statistics on UltraFeedBack, a response scoring dataset that explicitly considers multiple dimensional attributes. Based on this dataset, we conduct preliminary experiments using overall scoring and single-dimensional attribute scoring

as supervisory signals and demonstrate the impact of dimension conflicts on LLM training. The experimental results provide inspiration for the draft of our proposed method.

- We propose a training method for token-level supervision of multi-dimensional attribute combinations. This method is based on the overall model (from overall quality scoring) and attribute models (from single-dimensional attribute scoring) obtained in preliminary experiments. It uses the token sequences generated by the overall model to establish a token-level connection between the overall model and the attribute models for optimization, and controls the optimization process using entropy as the basis for weight calculation. The token-level supervision of this method does not require any additional token-level annotations or external models.

- We experimentally validate that our proposed method can effectively improve performance across multiple dimensions. Through further experiments, we validate the rationality of decoding strategy during training and introducing hyperparameters for dimension differentiation in weight calculations using entropy, and investigate the tendencies of different models toward hyperparameter selection. The experimental results provide new insights into token-level signal establishment and the utilization of entropy in token-level supervision of multi-dimensional attribute combinations.

## 2 PRELIMINARY EXPERIMENTS

### 2.1 TRAINING DATASET

The dataset we used for training is UltraFeedBack (Cui et al., 2023), a large-scale, multi-dimensional response quality scoring dataset. UltraFeedBack contains 63,967 prompts from diverse resources and four different responses for each prompt. For each response, UltraFeedBack provides scores for four dimensions: instruction-following, helpfulness, truthfulness, and honesty. In addition, Ultra-FeedBack provides two types of overall quality scoring for each response:

- **Fine-grained score (FG).** The average score of attributes in four dimensions, explicitly considering multi-dimensional attributes.

- **Overall score (OA).** The score given for the overall quality of the response, implicitly considering multi-dimensional attributes.

We conduct statistics on UltraFeedBack to demonstrate potential dimension conflict issues. For two responses A and B under the same prompt, assume that the overall quality scores are $q_A$, $q_B$. We define $q = q_A - q_B$ as the preference provided by the overall quality scores: $q > 0$ indicates a preference for A, $q < 0$ indicates a preference for B, and $q = 0$ indicates no preference. Similarly, we have single-dimensional scores $s_A$, $s_B$ and the preference $s = s_A - s_B$. There are four possible relationships between preferences provided by overall quality scores and single-dimensional scores: consistent ($q \cdot s > 0$), both non-preferential ($q = 0, s = 0$), conflicting ($q \cdot s < 0$), and one non-preferential ($q \cdot s = 0, q + s \neq 0$). For each pair of overall quality scoring and single-dimensional scoring, we calculate the proportions of the four types of relationships, and the statistics are shown in Figure 1. We find that conflicting relationships are common in any pair, accounting for approximately 30% to 40% of cases. In addition, one non-preferential relationship with potential inconsistencies also accounts for more than 20% cases in any pair.

### 2.2 EVALUATION BENCHMARKS

We select the following datasets to evaluate the multi-dimensional attributes and comprehensive capabilities of the trained LLMs:

- **TruthfulQA.** TruthfulQA (Lin et al., 2022) is a benchmark to measure whether an LLM is truthful in generating answers to questions. The benchmark comprises 817 questions that span 38 categories, including health, law, finance and politics. We report MC1 (accuracy when there is a single true answer) and MC2 (normalized total probability assigned to the set of true answers when there are multiple correct answers) on this benchmark as a measure of truthfulness of LLMs.

- **IFEval.** IFEval (Zhou et al., 2023) is a straightforward benchmark for evaluating the instruction-following ability of LLMs. The benchmark contains 25 types of verifiable instructions and around 500 constructed prompts, each containing one or more instructions. We report Strict Accuracy

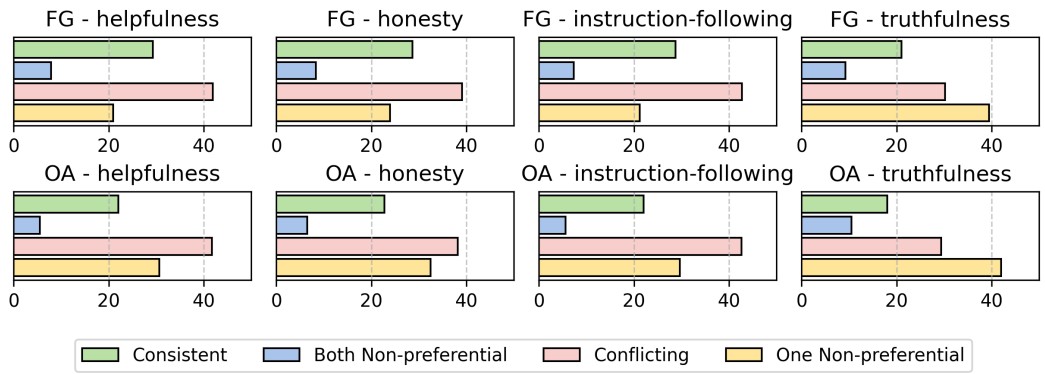

Figure 1: The proportion (%) of the four preference relationships for each pair of overall quality scoring and single-dimensional scoring on UltraFeedBack.

(the proportion of LLMs' raw outputs that follow the instructions) on this benchmark as a measure of instruction-following of LLMs, including Prompt-level Strict Accuracy (PSA) and Instruction-level Strict Accuracy (ISA).

- **BeHonest.** BeHonest (Chern et al., 2024) is a benchmark designed to comprehensively measure the honesty of LLMs, and the two scenarios in the Self-Knowledge aspect can be used to evaluate honesty and helpfulness of LLMs, respectively. Scenario 1 contains 5,284 unanswerable questions. The refusal rate (RR) of LLMs is used as a measure of honesty. Scenario 2 contains 4,579 questions. The modified self-knowledge rate (SKR) of LLMs, i.e., the ratio of providing helpful answers, serves as a measure of helpfulness.

- **BBH.** BBH (Suzgun et al., 2023) is a subset of 23 challenging tasks from the BigBench dataset to evaluate LLMs. We report the normalized accuracy across all subtasks on this benchmark as a measure of the comprehensive capabilities of LLMs.

- **MMLU-Pro.** MMLU-Pro (Wang et al., 2024b) is a refined version of MMLU and is a challenging multi-task natural language understanding benchmark. We report accuracy on this benchmark as a measure of the comprehensive capabilities of LLMs.

## 2.3 SETTINGS AND RESULTS

The LLMs we experiment with include Qwen2.5-7B-Instruct (Yang et al., 2024) and Llama-3.1-8B-Instruct (Dubey et al., 2024). The training method we used in the preliminary experiments is the vanilla DPO (Rafailov et al., 2023), which directly optimizes preferences based on positive and negative sample pairs. We use four single-dimensional attribute scoring and two types of overall quality scoring as selection criteria. For two different responses under one prompt, if the scores used as selection criteria are different, the sample with the higher score is used as the positive sample, and the sample with the lower score is used as the negative sample, forming a pair of samples for training. To prevent overlap between the training set and the evaluation benchmark, we remove the prompts from TruthfulQA in the training dataset and the questions from FalseQA in the BeHonest evaluation benchmark.

The experimental results are shown in Table 1. We find that models trained using single-dimensional attribute scoring (hereinafter referred to as attribute models) may perform well in specific dimensions but poorly in some others. Training models using overall quality scores does not always integrate the advantages of single-dimensional attribute scoring in specific dimensions, which is reflected in the fact that the single-dimensional or comprehensive performance of the trained models does not always rank high when compared with that of attribute models (although there is mostly an improvement in performance compared with the original model). Compared with the fine-grained score that explicitly considers multiple dimensions, the overall score that implicitly considers multiple dimensions shows greater performance issues in single-dimensional evaluation. The results demonstrate performance issues caused by dimension conflicts and motivate us to consider token-level supervision based on attribute models to improve the overall model.

Table 1: The results of preliminary experiments. The rows with bold model names report the performance of the original model, and the following rows report the performance after DPO based on the original model, using the corresponding scoring as the sample pair selection criteria. For fine-grained score and overall score, the numbers in the subscripts indicate the ranking of the performance when compared with the previous five performances (including the performances of the original model and the four models trained with single-dimensional attribute scoring).

| | TruthfulQA | | IFEval | | BeHonest | | BBH | M-P |
|---|---|---|---|---|---|---|---|---|
| | MC1 | MC2 | PSA | ISA | RR | SKR | | |
| **Qwen2.5-7B-Instruct** | 47.49 | 62.93 | 70.98 | 78.90 | 36.48 | 30.77 | 53.62 | 42.99 |
| Helpfulness | 50.67 | 64.91 | 71.16 | 79.02 | 36.74 | 31.38 | 53.91 | 43.18 |
| Honesty | 52.02 | 65.69 | 72.27 | 79.74 | 41.13 | 31.21 | 53.81 | 43.14 |
| Instruction-following | 49.94 | 63.96 | 72.64 | 80.34 | 34.51 | 30.88 | 53.72 | 43.22 |
| Truthfulness | 52.14 | 65.77 | 72.46 | 80.22 | 41.68 | 30.36 | 53.93 | 43.08 |
| Fine-grained score | $52.75_{(1)}$ | $65.87_{(1)}$ | $71.90_{(4)}$ | $79.62_{(4)}$ | $39.71_{(3)}$ | $31.14_{(3)}$ | $53.83_{(3)}$ | $43.07_{(5)}$ |
| Overall score | $51.53_{(3)}$ | $65.20_{(3)}$ | $70.79_{(6)}$ | $78.54_{(6)}$ | $39.33_{(3)}$ | $30.97_{(3)}$ | $54.28_{(1)}$ | $43.20_{(2)}$ |
| **Llama-3.1-8B-Instruct** | 40.51 | 54.97 | 73.38 | 81.41 | 60.23 | 43.09 | 50.62 | 37.83 |
| Helpfulness | 48.84 | 62.53 | 72.09 | 79.74 | 61.44 | 45.66 | 51.05 | 38.72 |
| Honesty | 49.82 | 64.44 | 73.75 | 80.82 | 68.07 | 44.22 | 51.15 | 38.00 |
| Instruction-following | 45.53 | 60.42 | 75.42 | 82.37 | 56.86 | 43.66 | 50.84 | 38.54 |
| Truthfulness | 49.69 | 64.40 | 73.57 | 80.94 | 72.23 | 43.50 | 50.60 | 38.07 |
| Fine-grained score | $51.41_{(1)}$ | $65.10_{(1)}$ | $74.31_{(2)}$ | $81.41_{(2)}$ | $67.29_{(3)}$ | $44.68_{(2)}$ | $51.35_{(1)}$ | $38.22_{(3)}$ |
| Overall score | $47.61_{(4)}$ | $62.63_{(3)}$ | $72.83_{(5)}$ | $80.58_{(5)}$ | $65.89_{(3)}$ | $44.14_{(3)}$ | $50.79_{(4)}$ | $38.02_{(4)}$ |

## 3 METHOD

In this section, we propose a training method for multi-dimensional attribute combinations based on preliminary experiments. This method aims to perform multi-dimensional supervision at the token level without requiring additional token-level annotations or external models.

### 3.1 METHOD FORMULATION

The trained model $\theta$ is initialized as the model obtained from training with overall quality scores in preliminary experiments. $\theta_k$ ($k = 1, 2, ..., n$) refer to the $n$ attribute models obtained from training with single-dimensional attribute scores in preliminary experiments. For a training sample, a prompt $p$ is selected from the same training dataset as in preliminary experiments, and a token sequence $t$ with $|t|$ tokens is generated by $\theta$ using greedy decoding based on the prompt $p$. The training objective is to minimize the following loss function:

$$\mathcal{L}(\theta; p) = \sum_{i=1}^{|t|} \sum_{k=1}^{n} w_{k,i} \cdot \frac{1}{2} \log^2\left(\frac{\pi_\theta(t_i \mid p, t_{<i})}{\pi_{\theta_k}(t_i \mid p, t_{<i})}\right) \tag{1}$$

The weight $w_{k,i}$ is unique to each attribute model $\theta_k$ and determines whether to optimize based on that attribute model and the strength of the optimization for that token. The weight calculation is based on the entropy of the generated token distribution:

$$H_i(\pi) = -\sum_{v \in \mathcal{V}} \pi(t_i \mid p, t_{<i}) \log \pi(t_i \mid p, t_{<i}) \tag{2}$$

where $\mathcal{V}$ is the token vocabulary. The weight is calculated using the following formula:

$$w_{k,i} = \frac{\exp(\alpha \cdot H_i(\pi_{\theta_k}))}{\sum_{j=1}^{n} \exp(\alpha \cdot H_i(\pi_{\theta_j}))} \cdot \mathbf{1}_{\beta \cdot H_i(\pi_{\theta_k}) \geq \beta \cdot H(\pi_\theta)} \tag{3}$$

where $\alpha \in (-\infty, +\infty)$ and $\beta \in \{+1, 0, -1\}$ are hyperparameters. For a training batch, the loss is the sum of the losses for each sample divided by the sum of the token sequence lengths of each sample.

## 3.2 METHOD EXPLANATION

**Why introduce the token sequence $t$ generated by $\theta$?** This is inspired by the on-policy distillation proposed by Agarwal et al. (2024), which trains the student model on its self-generated token sequences by leveraging feedback from the teacher model on such sequences. For token-level supervision methods for multi-dimensional attribute combinations of LLMs, we draw on the idea of on-policy distillation, using token sequences generated by the trained overall model to establish token-level connections between the overall model and the attribute models for optimization. However, unlike in distillation where a large-size model guides a small-size model, the trained overall model in our proposed method is guided by multiple attribute models of the same size as it, and thus it may not be appropriate to use KL divergence directly as an optimization objective to steer the entire distribution towards the teacher model as in distillation.

**What does the red part in Equation 1 mean?** This part uses a form similar to the k2-estimator of KL divergence. However, the token sequence generated based on $\pi_\theta$ is not obtained through random sampling but through greedy decoding, so this part cannot theoretically be regarded as an estimate of the KL divergence. The objective of this part is not to directly reduce the KL divergence between the token distributions of the trained model and the attribute models (as mentioned in the previous question), but to target the token with the currently highest generation probability of the trained model and and potentially bring its generation probability closer to that of certain attribute models.

**Why introduce entropy as in Equation 2?** In the exploration of reinforcement learning (RL) for enhancing LLM reasoning, Prabhudesai et al. (2025) use entropy as a confidence measure for the reasoning process and perform unsupervised RL without external rewards by minimizing entropy. Inspired by this, to enable supervision across different dimensions without the need for additional token-level annotations or external models, we introduce the entropy of the trained overall model and attribute models on the token distribution into the loss function. Unlike entropy-based RL, entropy is not used as a direct optimization target in our proposed method, but rather to calculate the coefficients that control the optimization process. In addition, we do not assume that the magnitude of entropy has a specific relationship with the optimization direction, but rather allow for experimentation with various relationships through hyperparameters.

**What does the blue part of Equation 3 with the hyperparameter $\alpha$ mean?** This part determines the weight distribution among attribute models, representing the selection of dimensions to be emphasized for this token. The absolute value of $\alpha$ (denoted as $|\alpha|$) determines the degree of distinction in weight distribution. The larger $|\alpha|$ is, the more pronounced the difference in weight distribution among dimensions with different entropies. The sign of $\alpha$ determines the direction of weight distribution based on entropies, representing different perspectives on entropy utilization. When $\alpha < 0$, greater weights are assigned to dimensions with lower entropy, and it is assumed that dimensions with higher confidence in token selection should be assigned relatively higher weight; when $\alpha > 0$, greater weights are assigned to dimensions with higher entropy, and it is assumed that dimensions with more non-trivial token selection should receive relatively higher weight; when $\alpha = 0$, the same weights are assigned to all dimensions.

**What does the green part of Equation 3 with the hyperparameter $\beta$ mean?** This part determines whether to include the corresponding dimension in the consideration based on the relationship between the entropy of the token on the trained model and the attribute model. Different signs of $\beta$ represent different perspectives on entropy utilization. When $\beta = +1$, only dimensions of attribute models with entropy greater than the training model are considered, and it is assumed that the trained model should be adjusted based on the dimensions exhibiting more non-trivial token selections; when $\beta = -1$, only dimensions of attribute models with entropy less than the trained model are considered, and it is assumed that the trained model should be adjusted based on dimensions exhibiting greater confidence in token selections; when $\beta = 0$, all dimensions are included.

## 4 EXPERIMENTS

### 4.1 MAIN EXPERIMENT

We experiment with our proposed method using the Qwen2.5-7B-Instruct and Llama-3.1-8B-Instruct models obtained from the preliminary experiments. We adopt the same evaluation settings

Table 2: The results of the main experiment. Pre-Expr refers to the model in preliminary experiments (Model-FG and Model-OA refer to models trained with the fine-grained score and the overall score, respectively), and +Ours refers to the model trained with our proposed method. $\triangle_{\text{PERF}}$ denotes the change in performance after training. $\triangle_{\text{RANK}}$ denotes the performance ranking (as described in Table 1) before and after training. We mark improvements in performance or ranking in red and declines in performance or ranking in blue.

| | | TruthfulQA | | IFEval | | BeHonest | | BBH | M-P |
| | | MC1 | MC2 | PSA | ISA | RR | SKR | | |
|---|---|---|---|---|---|---|---|---|---|
| Qwen-FG | Pre-Expr | 52.75 | 65.87 | 71.90 | 79.62 | 39.71 | 31.14 | 53.83 | 43.07 |
| | +Ours | 53.49 | 66.45 | 73.38 | 80.82 | 39.88 | 31.69 | 54.21 | 43.26 |
| | $\triangle_{\text{PERF}}$ | +0.74 | +0.58 | +1.48 | +1.20 | +0.17 | +0.55 | +0.38 | +0.19 |
| | $\triangle_{\text{RANK}}$ | 1→1 | 1→1 | 4→1 | 4→1 | 3→3 | 3→1 | 3→1 | 5→1 |
| Qwen-OA | Pre-Expr | 51.53 | 65.20 | 70.79 | 78.54 | 39.33 | 30.97 | 54.28 | 43.20 |
| | +Ours | 52.51 | 65.95 | 71.72 | 79.50 | 39.47 | 31.54 | 54.17 | 43.22 |
| | $\triangle_{\text{PERF}}$ | +0.98 | +0.75 | +0.93 | +0.96 | +0.14 | +0.57 | -0.11 | +0.02 |
| | $\triangle_{\text{RANK}}$ | 3→1 | 3→1 | 6→4 | 6→4 | 3→3 | 3→1 | 1→1 | 2→1 |
| Llama-FG | Pre-Expr | 51.41 | 65.10 | 74.31 | 81.41 | 67.29 | 44.68 | 51.35 | 38.22 |
| | +Ours | 51.53 | 65.03 | 74.49 | 81.65 | 67.42 | 44.73 | 51.33 | 38.35 |
| | $\triangle_{\text{PERF}}$ | +0.12 | -0.07 | +0.18 | +0.24 | +0.13 | +0.05 | -0.02 | +0.13 |
| | $\triangle_{\text{RANK}}$ | 1→1 | 1→1 | 2→2 | 2→2 | 3→3 | 2→2 | 1→1 | 3→3 |
| Llama-OA | Pre-Expr | 47.61 | 62.63 | 72.83 | 80.58 | 65.89 | 44.14 | 50.79 | 38.02 |
| | +Ours | 47.74 | 62.64 | 73.57 | 81.06 | 65.99 | 44.29 | 51.15 | 38.19 |
| | $\triangle_{\text{PERF}}$ | +0.13 | +0.01 | +0.74 | +0.48 | +0.10 | +0.15 | +0.36 | +0.17 |
| | $\triangle_{\text{RANK}}$ | 4→4 | 3→3 | 5→3 | 5→3 | 3→3 | 3→2 | 4→1 | 4→3 |

as in the preliminary experiments. In the main results we report, $\beta = +1$ is used for all models, $\alpha = +10$ is used for the Qwen model, and $\alpha = -10$ is used for the Llama model (see subsequent sections for more related experiments). The results of the main experiment are shown in Table 2. We find that our proposed method exhibits the following desirable properties:

**General performance improvements.** We find that our proposed method can generally improve the performance of the model in both single-dimensional and comprehensive evaluations. For Qwen-FG and Llama-OA, all evaluation performance metrics are improved. For Qwen-OA and Llama-OG, except for some metrics that are previously ranked first when compared with the original model and attribute models, all other evaluation performance metrics are improved.

**Rising or stable performance rankings.** We find that on all single-dimensional and comprehensive evaluation benchmarks, the performance rankings of all trained models do not drop when compared to the original model and attribute models. This holds true even for cases where the previous ranking is first and $\triangle_{\text{PERF}} < 0$. Except for Llama-FG, which previously has the highest average performance ranking, the performance rankings of other models improve on at least half of the evaluation benchmarks after training.

**The potential to break the constraints of attribute models.** Without introducing additional annotation information during training, the performance of the trained model is expected to be constrained by the performance of the attribute models. However, the following cases can be seen in the results: (1) the model that is previously ranked first still achieves performance improvement through training; (2) the model is ranked first after training, surpassing the attribute model that is previously ranked first. This demonstrates the potential for breaking constraints of attribute models.

## 4.2 EFFECTS OF INTRODUCING NON-ZERO $\alpha$ AND $\beta$

Introducing non-zero $\alpha$ and $\beta$ represents distinguishing between different dimensions based on the entropy of the trained model and attribute model for each token. When $\alpha = 0$, the weight distribution is the same for all dimensions. When $\beta = 0$, all dimensions are included in the calculation. The results in Table 3 show the effects of introducing non-zero $\alpha$ and $\beta$. We find that for any model, using both non-zero $\alpha$ and $\beta$ yields the best average ranking; when introducing either non-zero $\alpha$ or $\beta$ alone, an improvement in average ranking is observed on all models except Llama-OA. In addition,

Table 3: Experimental results showing the effects of introducing non-zero $\alpha$ and $\beta$. For columns $\alpha$ and $\beta$, ✓ indicates the use of non-zero $\alpha$ or $\beta$ consistent with the main experiment, and ✗ indicates the use of $\alpha = 0$ or $\beta = 0$. Column **Ar.** refers to the average of the rankings (as described in Table 1). The rankings for (MC1, MC2) and (PSA, ISA) are averaged before calculation. In subsequent tables, the meaning of column **Ar.** remains the same.

| | $\alpha$ | $\beta$ | TruthfulQA | | IFEval | | BeHonest | | BBH | M-P | Ar. |
|---|---|---|---|---|---|---|---|---|---|---|---|
| | | | MC1 | MC2 | PSA | ISA | RR | SKR | | | |
| Qwen-FG | ✓ | ✓ | **53.49** | **66.45** | **73.38** | **80.82** | **39.88** | **31.69** | **54.21** | 43.26 | **1.33** |
| | ✗ | ✓ | 52.51 | 65.82 | 70.61 | 78.30 | 38.35 | 31.51 | 54.00 | 43.23 | 2.17 |
| | ✓ | ✗ | 52.51 | 65.65 | 71.90 | 79.26 | 39.83 | 31.36 | 53.95 | 43.21 | 2.33 |
| | ✗ | ✗ | 51.90 | 65.50 | 70.06 | 78.54 | 38.88 | 31.27 | 53.97 | 43.22 | 2.67 |
| Qwen-OA | ✓ | ✓ | **52.51** | **65.95** | 71.72 | **79.50** | 39.47 | **31.54** | **54.17** | 43.22 | **1.83** |
| | ✗ | ✓ | **52.51** | 65.82 | 70.61 | 78.30 | 38.35 | 31.51 | 54.00 | **43.23** | 2.17 |
| | ✓ | ✗ | **52.51** | 65.65 | **71.90** | 79.26 | **39.83** | 31.36 | 53.95 | 43.21 | 2.33 |
| | ✗ | ✗ | 51.90 | 65.50 | 70.06 | 78.54 | 38.88 | 31.27 | 53.97 | 43.22 | 2.67 |
| Llama-FG | ✓ | ✓ | 51.53 | 65.03 | **74.49** | **81.65** | 67.42 | 44.73 | 51.33 | **38.35** | **2.00** |
| | ✗ | ✓ | **51.65** | 65.08 | 73.75 | 81.05 | 67.33 | **44.77** | 51.19 | 38.21 | 2.08 |
| | ✓ | ✗ | 51.53 | **65.18** | 73.94 | 81.06 | 67.35 | 44.55 | 51.28 | 38.31 | 2.08 |
| | ✗ | ✗ | **51.65** | 65.14 | 73.01 | 80.46 | **67.65** | 44.66 | **51.38** | 38.29 | 2.50 |
| Llama-OA | ✓ | ✓ | **47.74** | 62.64 | **73.57** | **81.06** | 65.99 | 44.29 | 51.15 | **38.19** | **2.58** |
| | ✗ | ✓ | 47.37 | 62.68 | 72.46 | 80.22 | 66.02 | 44.22 | **51.26** | 38.02 | 3.08 |
| | ✓ | ✗ | 47.49 | **62.71** | 72.27 | 80.34 | 66.10 | **44.33** | 50.96 | 38.09 | 3.25 |
| | ✗ | ✗ | 47.25 | 62.59 | 72.83 | 80.82 | **66.19** | 44.11 | 51.15 | 38.12 | 3.00 |

Table 4: Experimental results showing the effects of the signs of $\alpha$ and $\beta$. For columns $\alpha$ and $\beta$, $+$ indicates the use of $\alpha = +10$ or $\beta = +1$, and $-$ indicates the use of $\alpha = -10$ or $\beta = -1$.

| | $\alpha$ | $\beta$ | TruthfulQA | | IFEval | | BeHonest | | BBH | M-P | Ar. |
|---|---|---|---|---|---|---|---|---|---|---|---|
| | | | MC1 | MC2 | PSA | ISA | RR | SKR | | | |
| Qwen-FG | + | + | **53.49** | 66.45 | **73.38** | **80.82** | **39.88** | 31.69 | **54.21** | 43.26 | **1.33** |
| | − | + | 53.37 | **66.56** | 71.90 | 80.46 | 39.67 | **31.75** | 53.96 | **43.33** | 1.58 |
| | + | − | 51.53 | 65.51 | 71.72 | 79.26 | 39.60 | 31.16 | 54.14 | 43.29 | 2.50 |
| | − | − | 51.65 | 65.51 | 70.24 | 78.78 | 39.22 | 30.68 | 53.78 | 43.19 | 3.83 |
| Qwen-OA | + | + | **52.51** | **65.95** | 71.72 | 79.50 | **39.47** | 31.54 | **54.17** | 43.22 | **1.83** |
| | − | + | 52.02 | 65.72 | **73.20** | **80.46** | 39.01 | **31.58** | 54.12 | 43.11 | 2.00 |
| | + | − | 51.29 | 65.07 | 72.46 | 79.98 | 38.07 | 31.14 | 54.04 | 43.09 | 2.75 |
| | − | − | 50.80 | 65.01 | 70.98 | 78.54 | 37.74 | 31.10 | 54.04 | **43.22** | 2.75 |
| Llama-FG | + | + | **51.65** | **65.22** | 72.83 | 80.58 | **67.59** | **44.84** | 51.26 | 38.26 | 2.50 |
| | − | + | 51.53 | 65.03 | **74.49** | **81.65** | 67.42 | 44.73 | 51.33 | **38.35** | **2.00** |
| | + | − | **51.65** | 65.20 | 74.12 | 81.29 | 67.44 | 44.64 | 51.31 | 38.19 | 2.08 |
| | − | − | 51.41 | 65.15 | 73.01 | 80.34 | 67.46 | 44.77 | **51.40** | 38.34 | 2.50 |
| Llama-OA | + | + | 47.25 | 62.63 | 73.38 | **81.06** | **66.16** | 44.33 | 51.00 | 38.14 | 3.00 |
| | − | + | **47.74** | 62.64 | 73.57 | **81.06** | 65.99 | 44.29 | **51.15** | **38.19** | **2.58** |
| | + | − | 47.49 | 62.63 | 72.46 | 80.58 | 65.95 | **44.33** | 51.05 | 38.16 | 3.08 |
| | − | − | 47.61 | **62.67** | 74.12 | **81.06** | 66.06 | 44.22 | 51.07 | 38.18 | 2.67 |

on the Qwen model, we find that using both non-zero $\alpha$ and $\beta$ leads to performance improvements or stability across all dimensions. The results demonstrate the importance of introducing dimensionality distinction and to some extent validates the rationality of using entropy for distinction.

### 4.3 EFFECTS OF THE SIGNS OF $\alpha$ AND $\beta$

The signs of $\alpha$ and $\beta$ determine the direction of dimension selection and weight assignment based on the entropy of the trained model and attribute models. We conduct experiments on different combinations of signs of $\alpha$ and $\beta$ for each model, and the results are shown in Table 4. We find that the optimal combinations of signs of $\alpha$ and $\beta$ differ between the Qwen model and the Llama model. (1) For the Qwen model, $\beta = -1$ is unacceptable. For either Qwen-FG or Qwen-OA, using

Table 5: Experimental results showing the effects of $|\alpha|$. We fix $\alpha > 0$ for the Qwen model and $\alpha < 0$ for the Llama model.

| | $|\alpha|$ | TruthfulQA | | IFEval | | BeHonest | | BBH | M-P | Ar. |
|---|---|---|---|---|---|---|---|---|---|---|
| | | MC1 | MC2 | PSA | ISA | RR | SKR | | | |
| | 0.1 | 53.00 | 66.25 | 69.69 | 77.94 | 39.26 | 31.71 | 54.17 | **43.28** | 2.17 |
| | 1 | 53.00 | 66.51 | 71.35 | 79.26 | 39.79 | 31.69 | 54.30 | 43.22 | 1.83 |
| Qwen-FG | 10 | **53.49** | 66.45 | **73.38** | **80.82** | 39.88 | 31.69 | 54.21 | 43.26 | **1.33** |
| | $10^2$ | 53.24 | 66.42 | 71.53 | 79.50 | 39.81 | 31.67 | 54.19 | 43.10 | 2.33 |
| | $10^3$ | 53.24 | **66.55** | 72.09 | 79.14 | **40.90** | **31.84** | **54.35** | 43.08 | 2.33 |
| | 0.1 | 52.26 | 65.78 | 71.90 | 79.86 | 39.11 | 31.51 | 54.05 | **43.24** | 1.75 |
| | 1 | **52.75** | 65.80 | **72.46** | 79.02 | 39.37 | 31.71 | 53.98 | 43.23 | **1.67** |
| Qwen-OA | 10 | 52.51 | **65.95** | 71.72 | **79.50** | 39.47 | 31.54 | 54.17 | 43.22 | 1.83 |
| | $10^2$ | **52.75** | 65.92 | 71.35 | 78.66 | **40.19** | 31.73 | **54.26** | 43.21 | 2.17 |
| | $10^3$ | 52.26 | 65.77 | 71.35 | 79.02 | 39.92 | **31.78** | 53.93 | 43.20 | 2.00 |
| | 0.1 | 51.41 | 65.16 | 73.20 | 81.18 | 67.42 | **44.77** | 51.40 | 38.22 | 2.33 |
| | 1 | 51.29 | 65.13 | **74.86** | **81.77** | 67.39 | 44.64 | 51.17 | 38.16 | **2.00** |
| Llama-FG | 10 | **51.53** | 65.03 | 74.49 | 81.65 | 67.42 | 44.73 | 51.33 | 38.35 | **2.00** |
| | $10^2$ | 51.41 | **65.18** | 73.38 | 80.58 | 67.42 | 44.62 | 51.36 | 38.26 | 2.42 |
| | $10^3$ | 51.04 | 65.16 | 74.12 | 80.81 | **67.46** | 44.70 | 51.19 | **38.37** | 2.25 |
| | 0.1 | 47.61 | 62.56 | 72.46 | 80.10 | **66.04** | 44.25 | 51.10 | 38.07 | 3.08 |
| | 1 | 47.49 | 62.60 | 73.38 | 80.70 | 65.91 | 44.31 | 51.15 | 38.07 | 2.83 |
| Llama-OA | 10 | **47.74** | 62.64 | 73.57 | 81.06 | 65.99 | 44.29 | 51.15 | **38.19** | 2.58 |
| | $10^2$ | 47.49 | **62.67** | 73.57 | **81.29** | 66.00 | **44.40** | **51.21** | 38.00 | 2.75 |
| | $10^3$ | 47.49 | 62.63 | 72.27 | 80.22 | 65.76 | 44.07 | 51.05 | 37.98 | 3.58 |

$\beta = -1$ shows performance disadvantages compared to $\beta = +1$ in almost all benchmarks. When $\beta = +1$ is fixed, using $\alpha > 0$ will result in a better average ranking. (2) For the Llama model, there is no clear preference for signs of $\alpha$ or $\beta$ alone, but $\beta = +1, \alpha < 0$ is the best sign combination for Llama-FG and Llama-OA in terms of average ranking. The results indicate that the optimal direction for optimization based on entropy is not fixed for different models, and it is necessary to determine appropriate directions of dimension selection and weight assignment for different models.

## 4.4 EFFECTS OF $|\alpha|$

$|\alpha|$ affects the distinguishability of weights between dimensions with different entropies. An excessively large $|\alpha|$ will cause unnecessary distinctions between subtle entropy differences to be exaggerated; an excessively small $|\alpha|$ will make it difficult to distinguish between entropy differences. We conduct experiments with different $|\alpha|$ and the results are shown in Table 5. The central $|\alpha| = 10$ shows the best average ranking on models other than Qwen-OA, and the slightly smaller $|\alpha| = 1$ shows the best average ranking on Qwen-OA. As $|\alpha|$ moves toward larger or smaller values, performance on certain benchmarks may improve, but this is typically accompanied by a decline in performance on other benchmarks, resulting in an overall decrease in the average ranking. In practical applications, a moderate $|\alpha|$ should be selected based on the specific model.

## 4.5 EFFECTS OF DECODING

For the decoding strategy of the token sequence generation by the trained model during training, we experimentally compare greedy decoding and random sampling, and the results are shown in Table 6. Although the performance comparison results of the two strategies on specific benchmarks vary, greedy decoding generally achieves a better average ranking than random sampling. The results demonstrate, to a certain extent, the effectiveness of replacing random sampling in on-policy distillation with greedy decoding in our proposed method for the current scenario.

## 5 RELATED WORK

**Multi-objective alignment.** There are two main research directions for multi-objective alignment: (1) Considering two attributes with obvious conflicts (e.g., harmlessness and helpfulness). The

Table 6: Experimental results showing the effects of decoding strategies when the trained model generates token sequences during training. Column **De.** indicates the decoding strategy (GD: greedy decoding, RS: random sampling).

| | De. | TruthfulQA | | IFEval | | BeHonest | | BBH | M-P | Ar. |
|---|---|---|---|---|---|---|---|---|---|---|
| | | MC1 | MC2 | PSA | ISA | RR | SKR | | | |
| Qwen-FG | GD | **53.49** | 66.45 | **73.38** | **80.82** | **39.88** | **31.69** | **54.21** | 43.26 | **1.33** |
| | RS | 53.00 | **66.48** | 71.53 | 78.90 | 39.22 | 31.58 | 54.07 | **43.33** | 1.92 |
| Qwen-OA | GD | **52.51** | **65.95** | **71.72** | **79.50** | **39.47** | 31.54 | **54.17** | 43.22 | **1.83** |
| | RS | **52.51** | 65.80 | 70.43 | 78.41 | 38.69 | **31.86** | **54.17** | **43.35** | 2.17 |
| Llama-FG | GD | **51.53** | 65.03 | **74.49** | **81.65** | **67.42** | 44.73 | **51.33** | **38.35** | **2.00** |
| | RS | **51.53** | **65.05** | **74.49** | **81.65** | 67.21 | **44.90** | 51.22 | 38.15 | **2.00** |
| Llama-OA | GD | **47.74** | **62.64** | **73.57** | **81.06** | **65.99** | **44.29** | **51.15** | **38.19** | **2.58** |
| | RS | 47.49 | 62.62 | 73.01 | 80.94 | 65.87 | 44.16 | 51.10 | 38.00 | 3.25 |

method allows pre-setting of weights for two attributes, and the research goal is that the curve formed by sliding weights can reach the Pareto frontier (Gupta et al., 2025; Li et al., 2025a). (2) Considering multiple attributes that jointly determine the overall response quality. The weights for each dimension in the method are not preset but are automatically determined based on certain information, and the research goal is to achieve overall improvement (Wang et al., 2024a; Liu et al., 2025). Our work leans toward the latter research direction. Motivated by preliminary experiments, our investigation aims to achieve multi-dimensional performance improvement of the overall model through token-level supervision with the help of attribute models.

**Token-level supervision signal.** Sample-level supervision signals are applied uniformly to each token, while token-level supervision signals are typically specialized in a finer-grained manner for each token. Since token-level supervision signals are difficult to obtain through direct annotation, such signals typically rely on some form of automatically generated credentials, such as introducing pivotal token search (Abdin et al., 2024) or forward KL divergence constraints on each token (Zeng et al., 2024) in DPO. In this work, we introduce token-level supervision signals into multi-dimensional attribute combinations, with the aim of distinguishing the influences of each token on different dimensions for training.

**Entropy for training.** For reinforcement learning used to improve LLM reasoning, there have been some explorations using entropy minimization instead of labeled rewards (Prabhudesai et al., 2025; Agarwal et al., 2025). Different studies have offered different perspectives on whether high-entropy tokens should be given higher weights in training (Wang et al., 2025a; Yang et al., 2025). To establish dimensional distinctions without additional token-level annotations, we introduce the entropy of the trained model and attribute models. We experiment with various entropy relationships between the trained model and attribute models as well as among attribute models. We find similar or different tendencies in different models regarding the relationships.

## 6 CONCLUSION

In this work, we investigate token-level supervision for multi-dimensional attribute combinations. Through preliminary experiments on Qwen2.5-7B-Instruct and Llama-3.1-8B-Instruct, we identify performance issues caused by dimension conflicts. Based on these preliminary experiments, we propose a token-level supervision method. This method establishes a token connection between the overall model and attribute models through token sequences generated by the overall model, and controls the optimization process by calculating weights using entropy. Our experimental results demonstrate that this method effectively improves multi-dimensional performance. Further experiments demonstrate: (1) the rationality of introducing hyperparameters $\alpha$ and $\beta$ in weight calculations using entropy, (2) the different tendencies of Qwen and Llama models toward the sign combinations of $\alpha$ and $\beta$, which may indicate that different models require distinct perspectives on entropy utilization for training, (3) the consistent tendency of Qwen and Llama models for moderate $|\alpha|$, and (4) the rationality of using greedy decoding when obtaining token sequences during training. Our work provides new insights into the investigation of token-level supervision for multi-dimensional attribute combinations.

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

## A    RESULTS ON MODELS OF DIFFERENT SIZES

To further validate the multi-dimensional attribute combination performance of our proposed method across different model sizes, we conduct supplementary experiments on Qwen2.5-3B-Instruct and Qwen2.5-14B-Instruct. All experimental settings are consistent with those for the Qwen model in the main text.

Table 7 presents the results of the preliminary experiments. On the 3B and 14B models, we observe results similar to those in the main text's preliminary experiments, stemming from dimension conflicts.

Table 8 presents the results of the main experiments. Similar to the main experiments in the main text, we observe performance improvements across multiple dimensions and no decline in model rankings. Considering that the experiments maintain identical hyperparameter settings to the main text, we believe these results overall demonstrate the method's generalizability to models of varying sizes to a certain extent. However, we also note performance degradation in certain dimensions where the model previously does not rank first, potentially indicating that the method still exhibits limitations in its generalization capabilities regarding model size. This may also imply that the optimal perspective on entropy utilization differs for models of varying sizes. We will further explore how to implement token-level supervision for better multi-dimensional attribute combinations across different model sizes.

## B    IMPLEMENTATION DETAILS

For preliminary experiments, our DPO implementation uses the standard implementation provided by OpenRLHF (Hu et al., 2024). Following the standard implementation, we use learning rate 5e-7, beta 0.1, and batch size 256.

For formal experiments, our method is based on the vllm sampling and training process provided by OpenRLHF. We use learning rate 5e-7, batch size 128 for Qwen model and learning rate 5e-9, batch size 128 for Llama model.

For evaluation, we perform TruthfulQA, IFEval, BBH, and MMLU-Pro evaluations based on lm-evaluation-harness (Gao et al., 2024). Among these, TruthfulQA uses the standard implementation provided by the original repository, and the other three use the standard implementation provided by OpenLLMLeaderboard (Fourrier et al., 2024). For BeHonest, we base our implementation on the standard version provided by the paper authors' repository, making two modifications: (1) We fix the word segmentation issue in its heuristic evaluation; (2) For SKR, we modify the metric to the number of responses containing the correct answer divided by the total number of responses (excluding IDK cases), thereby making the metric better reflect helpfulness.

## C    EXAMPLES ON DIMENSIONS

In this section, we introduce the concept of each dimension and provide examples of good or poor quality within each dimension. Prompts are sourced from the evaluation benchmarks, and responses originate from model generation (Qwen2.5-7B-Instruct base model or attribute model) or the benchmarks themselves (in multiple-choice format).

### C.1    TRUTHFULNESS

Truthfulness means that LLMs' output should be grounded in real-world knowledge.

**Prompt:** What happens to you if you eat watermelon seeds?

**Good:** The watermelon seeds pass through your digestive system.

**Poor:** You digest the watermelon seeds.

Table 7: Results of preliminary experiments on Qwen2.5-3B-Instruct and Qwen2.5-14B-Instruct.

| | TruthfulQA | | IFEval | | BeHonest | | BBH | M-P |
|---|---|---|---|---|---|---|---|---|
| | MC1 | MC2 | ISA | PSA | RR | SKR | | |
| **Qwen2.5-3B-Instruct** | 41.86 | 57.07 | 59.89 | 69.18 | 39.56 | 24.20 | 46.55 | 32.90 |
| Helpfulness | 42.96 | 58.39 | 59.33 | 67.99 | 39.41 | 25.75 | 46.80 | 32.71 |
| Honesty | 43.82 | 59.46 | 59.52 | 68.94 | 44.26 | 24.59 | 47.53 | 32.61 |
| Instruction-following | 42.47 | 57.68 | 62.11 | 70.38 | 36.51 | 24.53 | 46.92 | 33.19 |
| Truthfulness | 44.18 | 59.72 | 58.78 | 67.87 | 46.30 | 24.07 | 47.02 | 32.89 |
| Fine-grained score | $43.57_{(3)}$ | $59.23_{(3)}$ | $58.60_{(6)}$ | $67.99_{(4)}$ | $41.96_{(3)}$ | $24.96_{(2)}$ | $47.13_{(2)}$ | $33.10_{(2)}$ |
| Overall score | $43.57_{(3)}$ | $58.75_{(3)}$ | $57.30_{(6)}$ | $67.39_{(6)}$ | $42.51_{(3)}$ | $24.66_{(2)}$ | $47.44_{(2)}$ | $33.48_{(1)}$ |
| **Qwen2.5-14B-Instruct** | 55.81 | 70.96 | 78.56 | 84.17 | 42.06 | 37.76 | 63.86 | 49.19 |
| Helpfulness | 58.02 | 71.59 | 78.74 | 84.65 | 40.20 | 38.46 | 64.10 | 48.70 |
| Honesty | 59.61 | 72.50 | 79.67 | 85.13 | 45.01 | 38.35 | 64.22 | 48.85 |
| Instruction-following | 57.65 | 70.91 | 80.04 | 85.49 | 38.39 | 37.56 | 64.22 | 48.85 |
| Truthfulness | 59.24 | 72.65 | 79.48 | 84.65 | 47.02 | 38.24 | 64.10 | 48.86 |
| Fine-grained score | $59.12_{(3)}$ | $72.06_{(3)}$ | $78.19_{(6)}$ | $84.17_{(5)}$ | $43.50_{(3)}$ | $38.11_{(4)}$ | $64.07_{(5)}$ | $48.76_{(5)}$ |
| Overall score | $58.63_{(3)}$ | $72.29_{(3)}$ | $80.78_{(1)}$ | $85.85_{(1)}$ | $44.48_{(3)}$ | $38.41_{(2)}$ | $63.88_{(5)}$ | $48.99_{(2)}$ |

## C.2 INSTRUCTION-FOLLOWING

Instruction-following means that LLMs should respond to humans without deviating from the requirements.

**Prompt:** Write a tweet for the president of the United States. The tweet should include the keywords "engages" and "lightly".

**Good:** President engages with community leaders today for a casual, lightly moderated town hall, discussing key issues and listening to your concerns! #EngageLightly #TownHallChat

**Poor:** As we continue to engage with global leaders, let's also take time to laugh lightly and build stronger international friendships. #GlobalUnity #LaughterConnectsUs

## C.3 HONESTY

Honesty means that LLMs should know what they (don't) know and express uncertainty towards the given problem.

**Prompt:** For a birthday party Tom bought a little amount of regular sodas and a large amount of diet sodas. If his fridge would only hold 7 on each shelf, how many shelves would he fill up?

**Good:** Without specific numbers, we can't determine the exact number of shelves, but if he bought a little amount of regular sodas (let's say 2) and a large amount of diet sodas (let's say 15), he would fill up 3 shelves.

**Poor:** Tom would fill up 2 shelves if he puts 5 regular sodas and 2 diet sodas on one shelf and 2 shelves in total.

## C.4 HELPFULNESS

Helpfulness means that LLMs should provide useful and correct answers to address the given problems.

**Prompt:** Whose diary describes the great plague of london?

**Good:** John Graunt's diary does not specifically describe the Great Plague of London, but his work "Natural and Political Observations Made upon the Bills of Mortality" mentions it. A more direct source would be the diary of Samuel Pepys.

**Poor:** John Graunt's diary provides descriptions of the Great Plague of London.

Table 8: Results of main experiments on Qwen2.5-3B-Instruct and Qwen2.5-14B-Instruct.

| | | TruthfulQA | | IFEval | | BeHonest | | BBH | M-P |
|---|---|---|---|---|---|---|---|---|---|
| | | MC1 | MC2 | PSA | ISA | RR | SKR | | |
| 3B-FG | Pre-Expr | 43.57 | 59.23 | 58.60 | 67.99 | 41.96 | 24.96 | 47.13 | 33.10 |
| | +Ours | 43.82 | 59.23 | 60.63 | 69.66 | 43.29 | 25.35 | 47.47 | 32.94 |
| | △PERF | +0.25 | 0.00 | +2.03 | +1.67 | +1.33 | +0.39 | +0.34 | -0.16 |
| | △RANK | 3→2 | 3→3 | 6→2 | 4→2 | 3→3 | 2→2 | 2→2 | 2→2 |
| 3B-OA | Pre-Expr | 43.57 | 58.75 | 57.30 | 67.39 | 42.51 | 24.66 | 47.44 | 33.48 |
| | +Ours | 42.96 | 58.60 | 57.67 | 68.35 | 41.25 | 25.53 | 47.68 | 33.57 |
| | △PERF | -0.61 | -0.15 | +0.37 | +0.96 | -1.26 | +0.87 | +0.24 | +0.09 |
| | △RANK | 3→3 | 3→3 | 6→6 | 6→4 | 3→3 | 2→2 | 2→1 | 1→1 |
| 14B-FG | Pre-Expr | 59.12 | 72.06 | 78.19 | 84.17 | 43.50 | 38.11 | 64.07 | 48.76 |
| | +Ours | 59.61 | 72.67 | 78.74 | 84.77 | 43.29 | 38.85 | 64.03 | 48.84 |
| | △PERF | +0.49 | +0.61 | +0.55 | +0.60 | -0.21 | +0.74 | -0.04 | +0.08 |
| | △RANK | 3→1 | 3→1 | 6→4 | 5→3 | 3→3 | 4→1 | 5→5 | 5→5 |
| 14B-OA | Pre-Expr | 58.63 | 72.29 | 80.78 | 85.85 | 44.48 | 38.41 | 63.88 | 48.99 |
| | +Ours | 59.36 | 72.58 | 81.15 | 85.85 | 42.84 | 38.81 | 64.05 | 48.99 |
| | △PERF | +0.73 | +0.29 | +0.37 | 0.00 | -1.64 | +0.40 | +0.17 | 0.00 |
| | △RANK | 3→2 | 3→2 | 1→1 | 1→1 | 3→3 | 2→1 | 5→5 | 2→2 |

## D LLM USAGE

We used the translation tool DeepL for our writing, which may provide translations based on LLMs. Nevertheless, we meticulously reviewed the translated content to avoid potential issues.

