# OpenReview forum: "Investigating Token-Level Supervision of Multi-Dimensional Attribute Combinations"
_ICLR.cc/2026/Conference — ICLR 2026 Conference Withdrawn Submission_

### Official Review · Reviewer_Vi5r · 2025-10-28

**Soundness:** 2
**Presentation:** 3
**Contribution:** 2
**Rating:** 4
**Confidence:** 3

**Summary:**

This work addresses dimension conflicts in multi-dimensional attribute combination training for LLMs by proposing an annotation-free token-level supervision method that establishes token connections between the overall and attribute models using generated token sequences and optimizes training through entropy-based weighting, with experimental results validating its effectiveness in improving multi-dimensional performance.

**Strengths:**

1. This method introduces token-level supervision for multi-dimensional attribute combinations without external annotations, which is novel.
2. The experiments it designed are sufficient and the text descriptions are detailed.
3. This method provides a new perspective on addressing dimension conflicts in multi-objective alignment.

**Weaknesses:**

1. The method is sensitive to α and β selections,  lacking theoretical guidance and how to choose suitable parameters is also a problem.
2. The applicability of this method to larger models or more dimensions is not fully explored.
3. The theoretical explanation of "entropy is not used as a direct optimization target in our proposed method, but rather to calculate the coefficients that control the optimization process" is not clear.

**Questions:**

1. The performance of different parameters varies on different datasets. How to determine the suitable ones?
2. Are there plans to extend the method to more dimensions or more models?
3. Please carefully check the weakness.

---

> ### Author Response · Authors · 2025-11-13
> **Response to Reviewer Vi5r**
>
> Thank you for your careful review and valuable suggestions. Here are our responses to your concerns.
>
> > The method is sensitive to α and β selections, lacking theoretical guidance and how to choose suitable parameters is also a problem. The performance of different parameters varies on different datasets. How to determine the suitable ones?
>
> We introduce α and β to determine weights based on entropy and find that different models may exhibit distinct orientations. However, we have not identified a method for determining the optimal parameters for a new model; instead, we can only identify them through experimentation.
>
> > The applicability of this method to larger models or more dimensions is not fully explored. Are there plans to extend the method to more dimensions or more models?
>
> We conduct preliminary explorations of models across different sizes and the results are shown in the appendix. We are supplementing experiments on Mistral-7B and will incorporate them into the paper for the next submission.
>
> Regarding the number of dimensions, UltraFeedBack already represents one of the datasets in the multi-attribute domain that includes a large number of evaluable dimensions (featuring 4 dimensions, whereas typical studies consider 2~3 dimensions).
>
> > The theoretical explanation of "entropy is not used as a direct optimization target in our proposed method, but rather to calculate the coefficients that control the optimization process" is not clear.
>
> In conventional entropy-based RL, the model's own entropy over tokens is directly used as the optimization objective. However, in the multi-dimensional attribute combinations we consider, the primary optimization goal is to align the probabilities of the comprehensive model and attribute models over tokens, rather than altering the comprehensive model's entropy over tokens. Entropy serves as the basis for calculating the weights of different attribute models.

---

> > ### Comment · Reviewer_Vi5r · 2025-11-22
> >
> > Thank you for the extended explanations. I decide to remain my score.

---

### Official Review · Reviewer_oSJA · 2025-10-29

**Soundness:** 2
**Presentation:** 2
**Contribution:** 2
**Rating:** 4
**Confidence:** 4

**Summary:**

This paper studies token-level supervision for multi-dimensional alignment in LLMs. Through statistical analysis of the UltraFeedback dataset, the authors demonstrate that 30-40% of response pairs exhibit conflicting preferences between overall quality scores and single-dimensional attribute scores. In a two-stage setup, they first obtain one overall and four attribute models via DPO, then connect the overall model to attribute models at the token level by training on the overall model's self-generated sequences; per-token, per-dimension entropy-based weights guide optimization without extra token-level annotations. Experiments on Qwen2.5-7B-Instruct and Llama-3.1-8B-Instruct across TruthfulQA, IFEval, BeHonest, BBH, and MMLU-Pro show general improvements in truthfulness, instruction following, honesty/helpfulness, and aggregate ability, with stable or rising rankings and indications that the method can surpass the best attribute model in some cases.

**Strengths:**

+ Originality: The paper systematically quantifies cross-dimension preference conflicts on UltraFeedback, finding 30–40% conflicts and >20% one-non-preferential cases, which clarifies why sample-level supervision can misalign token-wise effects. The paper introduces a token-level supervision scheme that uses self-generated sequences and entropy-based dimension routing ($\alpha$/$\beta$) to link an overall model with attribute models, eliminating the need for additional token-level annotations. This approach creatively combines on-policy distillation ideas with multi-objective alignment.

+ Quality: The preliminary experiments effectively motivate the proposed approach by demonstrating that overall-score training fails to integrate single-dimensional advantages. The main experiments evaluate across five diverse benchmarks (TruthfulQA, IFEval, BeHonest, BBH, MMLU-Pro), providing multifaceted assessment. The systematic ablation studies examining $\alpha$, $\beta$, $|\alpha|$, and decoding strategies demonstrate scientific rigor. Multi-scale experiments (3B-14B, Appendix A) demonstrate reasonable generalizability, though all use Qwen/Llama families, leaving transferability to other architectures (e.g., Mistral) unexplored.

+ Clarity: The paper is well structured, with a clear outline of the preliminary motivation, method and experiments. It provides an accessible taxonomy of four preference relations and a lucid explanation of why self-generated sequences and entropy are used. Appendix C provides examples of dimensions that illustrate the definitions.

+ Significance: This study provides a quantitative analysis of dimensional conflicts in multi-dimensional LLM alignment. It shows that 30–40% of the samples in the widely used UltraFeedback dataset exhibit such conflicts, which demonstrates the scale of the issue. The proposed method can enhance model performance without requiring additional token-level annotations, addressing the issue of the high costs of previous methods. The study reveals significant differences in optimal hyperparameters across various model families. While the paper does not offer an explanation for this phenomenon, it provides valuable insights for future research. Ablation studies provide empirical evidence for entropy-based weight allocation in LLM optimization.

**Weaknesses:**

1.	Only two model families, Qwen and LLaMA, are explored. The rationale behind the choice of models, training methods and parameter settings should be explained. For example, the optimal value of the hyperparameter $\alpha$ varies significantly between models: $\alpha = +10$ for Qwen models and $\alpha = -10$ for LLaMA models. The paper states that "it is necessary to determine appropriate directions of dimension selection and weight assignment for different models.", which implies that hyperparameter settings cannot be transferred between models. Therefore, extensive hyperparameter experiments must be conducted for each new type of model.
2.	While Appendix B provides some implementation details (learning rates, batch sizes, evaluation frameworks), it does not furnish code or pseudocode, rendering replication challenging. Furthermore, the authors ought to consider providing train/eval scripts, seeds, version pins, compute profile (GPU type, hours), and YAML configs.
3.	The red-highlighted part of Equation 1 "uses a form similar to the $k^2$-estimator of KL divergence", yet "this part cannot theoretically be regarded as an estimate of the KL divergence". The design lacks a theoretical basis and it is necessary to explain why this formula can achieve the stated objectives. It is suggested that the authors add ablation experiments, gradient analysis, and convergence/stability diagnostics related to forward/reverse KL divergence and top token cross-entropy variants to prove the rationality of the design.
4.	The performance improvement of many metrics is less than 1% (Table 2), and some metrics even show a decline (e.g., BBH metric of the Qwen-OA model: -0.11; MC2 metric of the Llama-FG model: -0.07, BBH metric of the Llama-FG model: -0.02). While the paper emphasizes that "all rankings improve or stabilize," the practical significance of such small gains is unclear.
5.	According to the description in the method section, all models should be saved during the training process. Specific details regarding memory usage and related conditions during training should be added in the appendix.
6.	Subsection 3.2 contains five lengthy Q&A-style paragraphs that could be presented in a table or list. There is no schematic diagram of the workflow method; an intuitive presentation of key information should be included.
7.	The paper cites methods such as AMOPO and gradient-adaptive policy optimization in the references and mentions them in the related work section. However, it does not make any experimental comparisons with these methods, making it difficult to evaluate the advantages and disadvantages of the proposed method. Furthermore, it lacks comparisons with mainstream approaches, and the experimental content should be enriched.
8.	Table 6 shows that greedy decoding outperforms sampling decoding in terms of average ranking, but the margin of advantage is extremely small (e.g., for the Llama-FG model: 2.00 vs 2.00). In some individual metrics, sampling decoding is actually superior. The existing evidence is insufficient to support the conclusion that "greedy decoding should be preferred."

**Questions:**

1.	How should practitioners determine optimal α and β for new models without extensive grid search? Is there a principled way to predict these based on model characteristics?
2.	Can you provide theoretical analysis explaining why the formulation in Eq. 1 should lead to better multi-dimensional performance, despite not being a proper KL divergence estimator?
3.	Please provide compute cost (FLOPs/throughput) and memory for training with n attribute models; compare to single objective DPO.
4.	The method's setup bears conceptual similarity to multi-objective optimization or even some model merging techniques. Can the authors clarify their method's novelty against existing MOO preference alignment frameworks and justify the decision to design this new entropy-based loss rather than adapting an existing MOO approach?
5.	Can you visualize which tokens get optimized toward which dimensions? Are there patterns?
6.	In the paper, different parameter settings are used for different LLMs, and only the two major model families, LLaMA and Qwen, are used in the paper. So, can this method be applied to other model series? Is it necessary to conduct a large number of experiments to explore the optimal hyperparameter settings for performance for any different LLM?
7.	Table 8 indicates that using the optimal 7B hyperparameters on the 14B model results in performance degradation on some metrics (e.g., RR: -1.64). Does this observation suggest that the hyperparameters must be re-tuned for every model scale, and if so, how does this affect the overall cost-benefit trade-off?
8.	Given the slight performance improvement of the method and the lack of multi-seed runs, I am concerned about statistical significance. Can the authors provide any other evidence of stability to increase confidence that these improvements are reproducible and not noise?
9.	The performance improvement in this study is rather modest. However, it seems necessary to load multiple models simultaneously during training, rather than a single one. Please justify why the multi-fold computational overhead is reasonable.

---

> ### Author Response · Authors · 2025-11-13
> **Response to Reviewer oSJA**
>
> Thank you for your careful review and valuable suggestions. We will make revisions based on your suggestions before submitting the paper to other conferences. Here, we provide an overall response to some points of your concerns.
>
> > α and β selections
>
> We introduce α and β to determine weights based on entropy and find that different models may exhibit distinct orientations. However, we have not identified a method for determining the optimal parameters for a new model; instead, we can only identify them through experimentation.
>
> > design of the Equation 1
>
> The red part of Equation 1 represents the k2 estimator form of the KL divergence. Under greedy decoding, its actual optimization objective is to align the probability of the maximum-likelihood token of the comprehensive model with that of the attribute model (since the minimum value of 0 is achieved when probabilities are equal). We observe that greedy decoding yields slightly better results than standard sampling, though no theoretical explanation exists. Considering this is not the primary motivation of our work, we will weaken the description of this part in the revision.
>
> > comparsions with baselines
>
> Despite extensive work on multi-objective optimization, we do not find work for token-level attribute model combinations that require no additional annotation to serve as a reasonable baseline. Therefore, our comparison primarily focuses on whether this method achieves improvements across multiple attributes compared to the results from the pre-experiment.
>
> > relatively slight improvement / decline
>
> Simultaneous improvement across multiple attributes with potential conflicts is quite challenging. Therefore, while improvement on a single objective may be relatively slight (and sometimes decline can be seen), the tendency to achieve simultaneous improvement across multiple attributes is non-trival. We will supplement with repeated experiments and conduct significance tests.
>
> > model family
>
> We are supplementing experiments on Mistral-7B and will incorporate them into the paper for the next submission.
>
> > overhead
>
> This method requires an average of 10 hours to train a 7-8B model on 8 A800 GPUs. In comparison, a single DPO takes an average of 4 hours to complete on 8 A800 GPUs.
>
> > practical application issues
>
> The primary distinction of our work compared to other MOO methods lies in token-level supervision as certain conflicts can only be mitigated by providing distinct optimization directions across different tokens. Our use of entropy stems from exploratory purposes. When determining the weights of individual attribute models at the token level, we struggle to identify a basis that does not require introducing additional annotations. Entropy stands as one of the few token-level bases available, prompting us to experiment with this signal.
>
> However, considering memory/time overhead and varying parameter biases across models, our approach faces practical challenges in implementation. With this in mind, we will shift our overall writing approach before the next submission, focusing more on investigating.

---

### Official Review · Reviewer_jBoU · 2025-10-31

**Soundness:** 2
**Presentation:** 1
**Contribution:** 1
**Rating:** 2
**Confidence:** 3

**Summary:**

This paper proposes a method to address the dimension conflict problem in large language models (LLMs). The approach aims to mitigate interference among multiple attribute dimensions during token-level supervision, while maintaining strong supervised performance.

**Strengths:**

- The overall idea is conceptually reasonable and could provide insights into improving the disentanglement and interpretability of LLM representations.

**Weaknesses:**

- **Poor writing and unclear motivation.**
The paper is difficult to follow, particularly in the introduction. The overall storyline, motivation, and problem formulation are not clearly articulated, making it hard to understand what specific challenge the paper aims to solve.

- **Lack of baseline comparison.**
It is unclear whether previous works have attempted similar solutions to dimension conflict. The paper does not provide direct comparisons with related approaches or meaningful baselines beyond simple variations of its own method.

- **Limited technical novelty.**
The proposed technique appears to be a minor variation on existing methods rather than a fundamentally new approach. The description of the method is overly brief, and the implementation details in the appendix are insufficient to fully understand the algorithm or reproduce the results.

**Questions:**

- Is it inherently difficult to make a direct comparison with other works that address dimension conflicts? If so, could the authors elaborate on the specific reasons (e.g., task formulation differences, lack of standardized benchmarks, or incompatible architectures)?

---

> ### Author Response · Authors · 2025-11-13
> **Response to Reviewer jBoU**
>
> Thank you for your careful review and valuable suggestions. Here are our responses to your concerns.
>
> > Poor writing and unclear motivation. The paper is difficult to follow, particularly in the introduction. The overall storyline, motivation, and problem formulation are not clearly articulated, making it hard to understand what specific challenge the paper aims to solve.
>
> We believe our motivation is clear: to explore how to use token-level supervision for multi-dimensional attribute combinations without requiring additional annotations. At the end of the Introduction, we have outlined the structure of the paper.
>
> > Lack of baseline comparison. It is unclear whether previous works have attempted similar solutions to dimension conflict. The paper does not provide direct comparisons with related approaches or meaningful baselines beyond simple variations of its own method. Is it inherently difficult to make a direct comparison with other works that address dimension conflicts? If so, could the authors elaborate on the specific reasons (e.g., task formulation differences, lack of standardized benchmarks, or incompatible architectures)?
>
> For multi-dimensional attribute combinations, we do not find work for token-level attribute model combinations that require no additional annotation to serve as a reasonable baseline. Therefore, our comparison primarily focuses on whether this method achieves improvements across multiple attributes compared to the results from the pre-experiment.
>
> > Limited technical novelty. The proposed technique appears to be a minor variation on existing methods rather than a fundamentally new approach. The description of the method is overly brief, and the implementation details in the appendix are insufficient to fully understand the algorithm or reproduce the results.
>
> We believe that within the context of multi-dimensional attribute combinations, considering how to mitigate dimensionality conflicts using token-level supervision without requiring additional annotations and investigating the signal of entropy is inherently novel.

---

> > ### Comment · Reviewer_jBoU · 2025-11-24
> >
> > Thank you for the rebuttal. However, I do not feel that my concerns have been sufficiently addressed, and in its current form, the paper does not meet the bar for acceptance at ICLR. My assessment appears to align with the other reviewers' perspectives as well. I will therefore keep my score as is.

---

### Official Review · Reviewer_BAbi · 2025-10-31

**Soundness:** 1
**Presentation:** 1
**Contribution:** 2
**Rating:** 2
**Confidence:** 2

**Summary:**

This paper investigates token-level supervision for multi-dimensional attribute training in LLMs. The authors provide preliminary analysis on UltraFeedback demonstrating that about 30% of preference pairs exhibit dimension conflicts, motivating the need for token-level rather than sample-level supervision. They propose a method that uses entropy-based dynamic weighting to optimize an overall model toward multiple attribute models without requiring token-level annotations. Experiments on Qwen and Llama models show performance improvements across multiple benchmarks, with the notable finding that different model families prefer opposite hyperparameter configurations.

**Strengths:**

- This study introduces a novel problem and appears to be the first trial of systematically investigating token-level supervision for multi-dimensional attribute conflicts in LLM alignment.
- This work takes an annotation-free approach that avoids the impractical requirement of token-level labels by leveraging entropy calculations from self-generated sequences and existing attribute models.

I acknowledge that my expertise in this specific research area is limited, and I might have over/understated the significance of this contribution.

**Weaknesses:**

Although I find the problem formulation interesting and the investigation of token-level supervision for multi-dimensional attributes to be a valuable contribution, I have some concerns about the mathematical presentation that made it challenging for me to fully understand the proposed method.

There are some unclear mathematical notions:
- In Equation 1, I found the notation $\log^{2}$ somewhat ambiguous. Could the authors clarify whether this means $\log \left[ \left( \frac{ \pi\_{\theta}(t_{i} | p, t_{<i} ) }{\pi_{\theta_{k}}(t_{i} | p, t_{<i})}  \right)^{2} \right]$ or $  \left[\log \left( \frac{ \pi\_{\theta}(t_{i} | p, t_{<i} ) }{\pi_{\theta_{k}}(t_{i} | p, t_{<i})} \right) \right]^{2}$?

- Additionally, I would appreciate some discussion about the rationale for using the squared term. It seems a simpler KLD-like form could serve a similar purpose, so understanding the motivation for this particular formulation would be helpful.

- The second paragraph of Section 3.2 describes what the red part of Equation 1 is not (i.e., not a KL divergence estimate), but I found it would be clearer if the authors could provide a more direct explanation of what this term is and why this particular formulation was chosen. The phrase "potentially bring its generation probability closer to that of certain attribute models" reads more as a hoped-for outcome rather than an explanation of how the loss function achieves this.

- Similarly, the third paragraph of Section 3.2 could benefit from additional clarification. While I understand the authors were inspired by Prabhudesai et al. (2025), the connection between their entropy-based RL approach and the use of entropy for weighting in this work could be explained more explicitly. The justification for why entropy (as opposed to other potential weighting signals) is an appropriate choice for this task would strengthen the work.

Some minor stuff:
- $\pi$, $t_{<i}$ are not defined.
- RHS of Eqn (2) is not indexed with $v$.
- $\mathbf{1}$ in Equation 3 needs to be defined.

These notation and explanation issues made it somewhat difficult for me to fully understand the implementation details of the proposed method. Since code has not been provided, additional clarity in the mathematical presentation would be particularly valuable. I believe addressing these concerns would significantly strengthen the paper, and I would be happy to reconsider my evaluation if these points can be clarified in a revision.

**Questions:**

- What is k2-estimator of KL divergence?
- I wonder if $\alpha$ could simply be explained as a sort of temperature scaling (although temperature scaling parameters are positive), which is a widely recognized concept.
- Are performance improvements significant? I'm less familiar with these datasets/evaluations, but they do not seem significant.

---

> ### Author Response · Authors · 2025-11-13
> **Response to Reviewer BAbi**
>
> Thank you for your careful review and valuable suggestions. Here are our responses to your concerns.
>
> > log^2, squared term and k2-estimator of KL divergence
>
> The squared term in Equation 1 is [log(...)]^2, i.e., first calculate the logarithm and then square it. Under standard sampling, this is known as k2-estimator of KL divergence. The k2 estimate is biased, but empirically exhibits minimal bias; moreover, it possesses low variance and is guaranteed to remain positive at all times. This property of guaranteed positivity makes it our primary candidate for consideration. We provided further explanation for this item under greedy decoding in our next reply.
>
> > The second paragraph of Section 3.2 describes what the red part of Equation 1 is not (i.e., not a KL divergence estimate), but I found it would be clearer if the authors could provide a more direct explanation of what this term is and why this particular formulation was chosen. The phrase "potentially bring its generation probability closer to that of certain attribute models" reads more as a hoped-for outcome rather than an explanation of how the loss function achieves this.
>
> Under greedy decoding, the current token is the one with the highest probability generated by the comprehensive model. For the squared term, its minimum value 0 is achieved when the probabilities of the comprehensive model and attribute model for that token are equal, so the optimization direction is to adjust the token probability to match the attribute model. We use "potentially bring" because the entropy-based weight is introduced, so this optimization may only occur with certain weights on specific attribute models.
>
> > Similarly, the third paragraph of Section 3.2 could benefit from additional clarification. While I understand the authors were inspired by Prabhudesai et al. (2025), the connection between their entropy-based RL approach and the use of entropy for weighting in this work could be explained more explicitly. The justification for why entropy (as opposed to other potential weighting signals) is an appropriate choice for this task would strengthen the work.
>
> Our use of entropy stems from exploratory purposes. When determining the weights of individual attribute models at the token level, we struggle to identify a basis that does not require introducing additional annotations. Entropy stands as one of the few token-level bases available, prompting us to experiment with this signal.
>
> > Some minor stuff:
>
> \pi denotes the model with corresponding parameters.
>
> \bm{t}_{<i} denotes the token sequence before index i.
>
> \bm{1} denotes 1 when the condition holds, and 0 otherwise.
>
> We mistakely use t_i instead of v in Eq. 2 and will correct this typo.
>
> > I wonder if \alpha could simply be explained as a sort of temperature scaling (although temperature scaling parameters are positive), which is a widely recognized concept.
>
> This coefficient is conceptually consistent with temperature, but it is used to adjust the degree of distinction based on entropy and possesses directionality.
>
> > Are performance improvements significant? I'm less familiar with these datasets/evaluations, but they do not seem significant.
>
> Simultaneous improvement across multiple attributes with potential conflicts is quite challenging. Therefore, while improvement on a single objective may be relatively slight, the tendency to achieve simultaneous improvement across multiple attributes is non-trival. We will supplement with repeated experiments and conduct significance tests.

---

### Note · Authors · 2025-12-02

I have read and agree with the venue's withdrawal policy on behalf of myself and my co-authors.